Effects of Tetranychus urticae infection on phyllosphere microbial community assembly of Vigna unguiculata

Chen DaWei 1
Xia GaoQin 1
Wang JiaoJiao 1
Luo YanYan 2
Wang HongLi nwnusky@163.com 2 3
Zhao Jing zhaojing@nwnu.edu.cn 1
Sun Kun 1
1 College of Life Sciences, The Northwest Normal University , Lanzhou , Gansu , China
2 Gansu Hospital of Traditional Chinese Medicine , Lanzhou , China
3 The Northwest Normal University , Lanzhou , Gansu , China
Mora-Montes Héctor
Electronic publication date: 2025 Dec 1
Publication date: 2025
Volume: 13
Electronic Location ID: e20389
Received 2025 Jul 29; Accepted 2025 Oct 24
Copyright: ©2025 Chen et al.
Copyright year: 2025
Copyright holder: Chen et al.
License: This is an open access article distributed under the terms of the Creative Commons Attribution License, which permits unrestricted use, distribution, reproduction and adaptation in any medium and for any purpose provided that it is properly attributed. For attribution, the original author(s), title, publication source (PeerJ) and either DOI or URL of the article must be cited.
License URL: https://creativecommons.org/licenses/by/4.0/

Keywords: Microbial assembly, Microbial network, Pest-microbe-plant interaction, Phyllosphere microbial taxa, Tetranychus urticae

Funding: Gansu science and technology plan project 25YFFA061 National Natural Science Foundation of China 82360745 Gansu Province Science and Technology Plan Project 23JRRA694 Research Ability Improvement Program for Young Teachers of Northwest Normal University NWNU-LKQN2023-20 Gansu Province Science and Technology Plan Project 24CXGA007 This research was supported by Gansu science and technology plan project under Grant No. 25YFFA061; National Natural Science Foundation of China under Grant No. 82360745; Gansu Province Science and Technology Plan Project under Grant No.23JRRA694; Research Ability Improvement Program for Young Teachers of Northwest Normal University under Grant No. NWNU-LKQN2023-20; Gansu Province Science and Technology Plan Project under Grant No.24CXGA007. The funders had no role in study design, data collection and analysis, decision to publish, or preparation of the manuscript.

==============================
Tetranychus urticae are among the most important leaf-damaging plant-pests, causing severe crop losses worldwide. The plant phyllosphere microbe plays fundamental roles in plant growth and health. However, little is known about how T. urticae and phyllosphere microbes interact to impact plant health. In this study, we used amplicon sequencing to explore the changes in phyllosphere microbes between infected and uninfected Vigna unguiculata leaves by T. urticae. The results showed that the diversity of epiphytic bacteria and endophytic fungi can be significantly decreased, influenced the community structure of the phyllosphere microbe, and decreased co-occurrence network connectivity and complexity of phyllosphere microbes after infection of T. urticae. After infection by T. urticae, V. unguiculata recruited some beneficial microbes (Rickettsia, Naganishia, Brevundimonas, and Aspergillus) to the phyllosphere. PICRUSt and FUNGuild predictive analysis indicated that infection of T. urticae can cause the changes of the function of the phyllosphere fungi. Null model analysis indicated that assembly of epiphytic and endophytic fungal community changed from deterministic processes to stochastic processes after infection of T. urticae, while assembly of epiphytic and endophytic bacterial community changed from stochastic processes to deterministic processes. Our findings provided new insights into interactions among phyllosphere microbes-pest-plants.

Introduction

The phyllosphere refers to the above-ground part of a plant, which contains different microorganisms in both epibiotic (organisms that reside on the surface of the leaf) and endophytic (organisms that live inside the leaf) niches (Xu et al., 2022). The microbial communities living in the phyllosphere environment are defined as phyllosphere microbiota (Remus-Emsermann & Schlechter, 2018). It is reported that the phyllosphere is the largest microbial habitat, covering a global area of about 108 km−2 (Kumar et al., 2023). Complex communities of diverse microorganisms inhabit various regions of the phyllosphere (Farré-Armengol et al., 2016). The surfaces of leaves are exposed to various biological stressors (such as insect infestations and pathogens) and abiotic stressors, thus harboring symbiotic, pathogenic, and beneficial microbial species. The same applies to the endosphere within plants (Liu, Brettell & Singh, 2020b). Phyllosphere microbiota critically mediate host ecological adaptation through development stimulation, nutrient acquisition, and abiotic/biotic stress mitigation (Stone, Weingarten & Jackson, 2018; Thapa & Prasanna, 2018; Xu et al., 2020; Chaudhary et al., 2021). The assembly of phyllosphere microbes is influenced by multiple biotic and abiotic factors (Vorholt, 2012). Pathogen infection is one of the most influential biological stresses affecting phyllosphere microbe assembly (Sun et al., 2023). Experimental findings demonstrate that plant species activate stress-mitigation mechanisms through the “cry for help” response. For example, when subjected to biotic or abiotic stressors, plants are capable of recruiting beneficial microbes from their surrounding environment (Cordovez et al., 2019). The mechanisms underlying plant-microbe interactions are regarded as essential for safeguarding plant health (Carrión et al., 2019). Rizaludin et al. (2021) identified that plants encounter a variety of biotic and abiotic stresses throughout their life cycle. In response to these stresses, plant roots can employ a “cry for help” mechanism, which facilitates the recruitment of beneficial microorganisms that assist in mitigating the detrimental effects of such stressors. Selective enrichment of rhizosphere microbiota in disease-resistant plant genotypes (e.g., tomato and common bean) correlates with pathogen suppression (Kwak et al., 2018; Mendes et al., 2018), establishing functional links between microbial consortia and host phenotypic traits. This finding suggests that particular soil microbes and their associated functions play a significant role in influencing plant phenotypes. Furthermore, when subjected to various plant pathogens, Arabidopsis thaliana selectively recruited specific beneficial microbes within its rhizosphere (Berendsen et al., 2018). This process is proposed to be mitigated by root exudates that are modified in response to pathogen infection (Yuan et al., 2018). These botanical responses may engender a legacy effect within the soil, thereby augmenting the survival rates of plant progeny and fostering the development of disease-suppressive soils (Berendsen et al., 2018; Yuan et al., 2018). Compared with the rhizosphere, the phyllosphere—another critical microbial colonization interface where plant hosts are exposed to the aerial environment—faces more direct aboveground biotic stressors. By analogy with the rhizosphere’s response pattern under pathogen stress, does the phyllosphere microbiome undergo similar community restructuring and diversity changes when the plant phyllosphere is subjected to biotic stress? Li et al. (2022) elucidated the differences in phyllosphere microbiomes between host leaves that were infected and those that remained uninfected by pathogens. They also identified potential mechanisms through which the observed shifts in the phyllosphere microbiome may have facilitated plant resilience against pathogen pressure.

The study of the phyllosphere microbe mainly focuses on the influence of pathogens, while neglecting other influences, especially pests which are seriously harmful to plants. Both pathogens and pests disrupt the structure and function of the phyllosphere microbial community in plants. Through distinct infection strategies, action mechanisms, and impacts on host physiology, they shape differential microbial community responses. Studies have shown that pathogens tend to invade primarily at the molecular level (Munzert & Engelsdorf, 2025), whereas pests mostly adopt physical damage or chemical induction for invasion (Bensoussan et al., 2016). After infecting plant hosts, pathogens directly inhibit antagonistic bacteria to form a pathogen-dominant community (Li et al., 2021), while pests indirectly alter the composition of microbial communities (Coolen et al., 2024). Then, what kind of impact does pest infestation actually have on the phyllosphere microbial community of host plants? Will it, just like the rhizosphere responds to stress, construct a defense system by specifically recruiting beneficial microorganisms? In-depth analysis of this scientific question can provide brand-new ideas and theoretical support for achieving green pest control through phyllosphere microbial regulation.

Tetranychus urticae is one of the most pernicious pests affecting crops and has a wide host range in the world, which can result in graying and yellowing, irregular shaped, white or gray necrotic spots on the leaves, and damage to greenhouse vegetables and fruits (Jakubowska et al., 2022). As a leguminous crop, cowpea (Vigna unguiculata) possesses a certain level of defense capability when encountering insect infestations. Recent studies have shown that when Vigna unguiculata is infested by Megalurothrips usitatus, the biosynthetic pathways of key defensive secondary metabolites such as flavonoids and phenylpropanoids are significantly upregulated, and compounds with known insect-resistant activity (e.g., naringenin) are accumulated (Li et al., 2023). This indicates that cowpea can activate a set of defense systems when facing insect infestation. It has been reported that the defense system of the leguminous plant Phaseolus lunatus is activated when infested by Tetranychus urticae (Li et al., 2025); therefore, we hypothesize that Tetranychus urticae feeding on leguminous plants triggers defense responses. However, the response of phyllosphere microbes to pest challenges encountered by plant hosts remains unknown. Therefore, investigating the impact of Tetranychus urticae on the microbial community of this plant with a well-defined defense mechanism can deeper reveal the interactive relationships among the “plant-pest-microbe” tripartite system. Accordingly, in this study, we used amplicon approaches to investigate the difference of phyllosphere microbial diversity, community composition and co-occurrence patterns between T. urticae-infected and uninfected leaves of Vigna unguiculata. These findings may establish a basis for enhancing our understanding of the relationships among pests, plants, and microbes.

Materials and Methods

Experimental materials

Two groups of leaf samples from V. unguiculata were collected in June 2023 at the Agricultural Pest Control Laboratory, College of Plant Protection, Gansu Agricultural University, located in Lanzhou (36°05′18″N, 103°42′16″E, 1,531 m), Gansu Province, China. The samples included T. urticae-infected leaves (designated as UH) and uninfected leaves (designated as H). HE, UHE represent endophytes in the uninfected and infected leaves of V. unguiculata, respectively; HA, UHA represent the epiphyte in the uninfected and infected leaves of V. unguiculata, respectively. Visual symptom assessment differentiated infected leaves from uninfected counterparts (Fig. S1). We collected 0.5 g of sample (infected/uninfected leaves) of V. unguiculata from each group as a sample, three biological replicates were selected. In this experiment, each replicate consisted of one individual plant. Three leaves were sampled from each plant and pooled together to form one biological sample. The experiment included three independent biological replicates (three separate plants), resulting in a total of nine leaves collected (three plants × three leaves per plant). Following 15 days of pest infestation, the host plants were sampled. Epiphytic microbiota was harvested by immersing foliar samples in 200 mL of sterile phosphate-buffered saline (0.01 M PBS, 4 °C) within 250 mL Erlenmeyer flasks, followed by sequential ultrasonic processing (15 min) and orbital agitation (200 rpm, 20 °C, 1 h) to dislodge surface-associated microorganisms into the liquid phase. Microbial biomass was harvested through vacuum filtration (0.22 µm cellulose membranes), with aseptic membrane dissection (2  × 2 mm fragments) performed using ethanol-sterilized instruments (Magare, Nair & Khairnar, 2017). In order to collect the endophyte, each leaf sample was surface sterilized following the previously described steps. This involved consecutive immersion in 75% ethanol for 1 min, followed by 3 min in 1% sodium hypochlorite, and a final immersion of 30 s in 75% ethanol. The samples were then rinsed three times with sterile water. Leaf samples underwent lyophilization followed by mechanical disruption post-treatment (Li et al., 2022). Processed specimens underwent cryopreservation at −80 °C prior to downstream analysis.

DNA extraction and amplicon sequencing

Genomic DNA isolation from surface-processed foliar specimens was performed using a commercial soil DNA extraction system (MP Biomedicals, Cat. No. 116560200). The integrity of the nucleic acids was assessed through agarose gel electrophoresis (1% w/v), while quantification was carried out via UV-spectrophotometric analysis, ensuring an absorbance ratio of A260/A280 ≥ 1.8.

The V5–V7 region of bacterial 16S rRNA gene (799F/1193R) (Li et al., 2024a) and the fungal ITS1 region (ITS1F/ITS2R) (Chen et al., 2021) were amplified. High-throughput sequencing was conducted on the NovaSeq 6000 platform (Illumina, San Diego, CA, USA) for library analysis.

Statistical analysis

Bioinformatic processing followed established QIIME workflows (Caporaso et al., 2010). Microorganism-derived sequences underwent quality trimming and were subsequently sorted into individual samples through barcode-based identification. Operational taxonomic units (OTUs) were classified through sequence alignment with the Silva bacterial 16S reference database (Magoč & Salzberg, 2011). OTUs were taxonomically resolved to species through reference-based alignment with the UNITE curated fungal database (Mejía et al., 2008). Sequence assembly into OTUs was conducted with a 97% sequence identity threshold through the USEARCH platform (http://drive5.com/uparse/) (Magoč & Salzberg, 2011). The OTU sequences of the samples were annotated using the QIIME software (Version 1.9.1) with the blast method as described by Caporaso et al. (2010) (http://qiime.org/scripts/assign_taxonomy.html). The analysis was conducted against the Unite database (version available at https://unite.ut.ee/index.php), allowing for a comprehensive examination of microbial community composition. QIIME (Version 1.9.1) was employed to assess the community diversity index of microbial communities, including indicators such as Chao 1 and Shannon (Kõljalg et al., 2013). Beta diversity patterns were visualized through NMDS ordination employing Bray-Curtis dissimilarity metrics across microbial assemblages (Knight et al., 2018). We quantified the relative importance of stochastic and deterministic processes in fungal community assembly using the taxonomic normalized stochasticity ratio (tNST) proposed by Ning et al. (2019). The tNST metric ranges from 0 to 1, with values >0.5 indicating dominance of stochastic processes and values <0.5 reflecting deterministic dominance. This method was implemented through the NST R package (version ≥ 3.0.3). The statistical analyses (including NST calculations and permutational ANOVA) were performed in R version 4.4.3. Final visualizations were generated using GraphPad Prism 10.1.2 (GraphPad Software, San Diego, CA, USA). Microbial co-occurrence networks were constructed based on species abundance data using correlation-based approaches (Berry & Widder, 2014; Friedman & Alm, 2012). Network inference was performed based on SparCC correlation analysis at the OTU level, with only those OTUs meeting an abundance threshold of >0.1% included in the analysis. Specifically, the number of connections was obtained from SparCC correlations. Additionally, node connectivity, representing the average number of connections per node in the network, was calculated based on these filtered and qualified connections. In accordance with established analytical workflows, the bacterial genomic prediction method integrated with functional annotation (PICRUSt) was employed, along with the fungal trait database (FUNGuild v1.0) (Langille et al., 2013; Nguyen et al., 2016).

Results

Alpha diversity

Alpha diversity indices analysis showed that after T. urticae infection, the endophytic bacteria Chao1 index was higher than that of the uninfected group (P > 0.05), while the endophytic bacteria shannon index was lower than that of the uninfected group (P > 0.05) (Fig. 1A). After T. urticae infection, Chao1 and Shannon indices of epiphytic bacteria and endophytic fungi decreased significantly (P < 0.05) (Figs. 1B, 1C). The Chao1 index of epiphytic fungi was higher than that of uninfected group (P > 0.05), and the Shannon index of surface epiphytic fungi was lower than that of uninfected group (P > 0.05) (Fig. 1D).

Figure 1 Alpha diversity changes of phyllosphere microbe.

(A) Endophytic bacteria, (B) epiphytic bacteria, (C) endophytic fungi, (D) epiphytic fungi. HE, UHE represent endophytes in the uninfected and infected leaves of V. unguiculata, respectively; HA, UHA represent the epiphyte in the uninfected and infected leaves of V. unguiculata, respectively.

Community composition

The bacterial OTUs were assigned into 13 phyla and 234 genera in all samples. The dominant bacterial phylum across all of the samples was Proteobacteria, with relative abundances ranging from 84.23% to 97.78% (Fig. 2A). At the genus level, Ralstonia was the dominant genus in the sample of HE (19.22%), Rickettsia was the dominant genus in the sample of UHE (59.15%), and Sphingomonas was the dominant genus in the sample of HA and UHA (40.68%, 32.20%, respectively) (Fig. 2B).

Figure 2 Relative abundances of phyllosphere microbe.

(A) Endophytic and epiphytic bacteria at the phylum level, (B) endophytic and epiphytic bacteria at the genus level, (C) endophytic and epiphytic fungi at the phylum level, (D) endophytic and epiphytic fungi at the genus level . “Other” represents the total of relative abundance outside the top ten maximum relative abundance levels. HE, UHE represent endophytes in the uninfected and infected leaves of V. unguiculata, respectively; HA, UHA represent the epiphyte in the uninfected and infected leaves of V. unguiculata, respectively.

The fungal OTUs were assigned to eight phyla and 117 genera, including epiphyte and endophyte. At the phylum level, Ascomycota was the dominant phylum in three samples (HE, HA, UHA), with relative abundances ranging from 37.21% to 90.41%. Basidiomycota was the dominant phylum in the UHE sample (92.30%) (Fig. 2C). At the genus level, Aureobasidium was the dominant genus in the HE sample (32.13%). Naganishia was the dominant genus in the sample of UHE (92.27%), and Simplicillium was the dominant genus in the sample of HA and UHA (15.23%, 16.04%, respectively) (Fig. 2D).

To better understand the pattern of microbial recruitment. Through analyzing the amplicon data, we found that T. urticae infection significantly affected the relative abundance of certain bacteria: the endophytic genus Rickettsia increased by 59.13%, while the epiphytic genera Rickettsia and Brevundimonas increased by 18.62% and 11.30%, respectively (Figs. 3A, 3B, Fig. S2A).

Figure 3 The impact of T. urticae infection on the abundance of major bacterial and fungal genera.

(A) Endophytic bacteria, (B) epiphytic bacteria, (C) endophytic fungi, (D) epiphytic fungi. Two-way ANOVA was used to compare the significant differences in treatments. Significant differences are indicated with asterisks above columns (*, p < 0.05). HE, UHE represent endophytes in the uninfected and infected leaves of V. unguiculata, respectively; HA, UHA represent the epiphyte in the uninfected and infected leaves of V. unguiculata, respectively.

After T. urticae infection, endophytic fungal genus (Naganishia, Aureobasidium, Malassezi a, Cladosporium, and Fusarium) were significantly affected. However, only the relative abundance of Naganishia was significantly increased, the relative abundance increased to 92.21%. While the epiphytic fungal genus (Aspergillus) was significantly affected, and the relative abundance increased by 22.53% (Figs. 3C, 3D, Fig. S2B).

Beta diversity

The results of NMDS showed that the phyllosphere bacterial and fungal community composition was generally separated when comparing infected and uninfected leaves (Figs. 4A, 4B), which implied that the infection of pest altered the bacterial and fungal community structure in leaves.

Biomarkers for infected and uninfected leaves of V. unguiculata at different samples

Linear discriminant biomarker analysis (LEfSe) revealed differences in the phyllosphere bacterial and fungal communities between infected and uninfected leaves of V. unguiculata at different samples. In the bacteria community, there were significant differences at the class (Gammaproteobacteria), the order (Burkholderiales, Enterobacterales, and Pseudomonadales), the family (Pseudomonadaceae), and the genus (Citrobacter and Pseudomonas) levels in the HE samples. There were significant differences at the class (Alphaproteobacteria), the order (Caulobacterales and Rickettsiales), the family (Caulobacteraceae and Rickettsiaceae), and the genus (Brevundimonas and Rickettsia) levels in the UHE samples. There were significant differences at the phylum (Actinobacteriota), the class (Actinobacteria), the order (Corynebacteriales and Sphingomonadales), the family (Dietziaceae, Beijerinckiaceae, and Sphingomonadaceae), and the genus (Dietzia, Methylobacterium -Methylorubrum, Bradyrhizobium, and Sphingomonas) levels in the HA samples. There were significant differences at the family (Bacillaceae), and the genus (Bacillus) levels in the UHA samples (Fig. 5A).

Figure 4 Non-metric multi-dimensional scaling (NMDS) ordinations of phyllosphere.

(A) Bacteria, (B) fungi. HE, UHE represent endophytes in the uninfected and infected leaves of V. unguiculata, respectively; HA, UHA represent the epiphyte in the uninfected and infected leaves of V. unguiculata, respectively.

Figure 5 The linear discriminant analysis effect size (LEfSe) analysis between infected and uninfected leaves of V. unguiculata at different samples.

(A) Phyllosphere bacterial, (B) phyllosphere fungi. The concentric circles radiating from the center represent different taxonomic levels. The yellow nodes indicate no significant difference. Only taxa with LDA > 4 and Wilcoxon, P < 0.05 are shown. HE, UHE represent endophytes in the uninfected and infected leaves of V. unguiculata, respectively; HA, UHA represent the epiphyte in the uninfected and infected leaves of V. unguiculata, respectively.

In the fungal community, there were significant differences at the class (Malasseziomycetes), the order (Dothideales and Malasseziales), the family (Saccotheciaceae and Malasseziaceae), and the genus (Aureobasidium and Malassezia) levels in the HE samples. There were significant differences at the phylum (Basidiomycota), the order (Filobasidiales), the family (Filobasidiaceae), and the genus (Naganishia) levels in the UHE samples. There were significant differences at the class (Dothideomycetes, Sordariomycetes, and Agaricomycetes), the order (Capnodiales, Pleosporales, and Hypocreales), the family (Cladosporiaceae, Didymosphaeriaceae, and Nectriaceae), and the genus (Cladosporium and Fusarium) levels in the HA samples. There were significant differences at the phylum (Ascomycota), the class (Eurotiomycetes), the order (Eurotiales and Microbotryomycetes-ord-incertae-sedis), the family (Pleosporaceae, Aspergillaceae, and Chrysozymaceae), and the genus (Albifimbria and Sampaiozyma) levels in the UHA samples (Fig. 5B).

Co-occurrence network analysis

To explore the effect of T. urticae infection on the co-occurrence pattern of phyllosphere microbes, we performed co-occurrence network analysis and estimated the topological properties to uncover potential associations and complexity of connections among the phyllosphere microbes. The results showed that co-occurrence network connectivity and complexity of phyllosphere microbes were declined after T. urticae infection compared with the uninfected condition (Fig. 6). Specifically, we observed that pest infestation disrupted the leaf microbiota, leading to a marked simplification of the bacterial-fungal network structure as evidenced by reduced connectivity and complexity (Figs. 6A, 6B, 6C, 6D). The topological properties (Table S1) quantified a marked simplification of the microbial network following pest infestation. Microbial networks in infected leaves exhibited a reduced average degree (bacteria: 7.54 vs. 16.67; fungi: 6.70 vs. 9.66) compared to uninfected leaves, indicating a less interconnected community.

Figure 6 Co-occurrence networks between infected and uninfected phyllosphere microbe.

Network inference was performed based on SparCC correlation at OTUs level (abundance > 0.1%) to compare the bacterial community interactions between (A) infected bacteria, (B) uninfected bacteria, (C) infected fungi, (D) uninfected fungi.

Function prediction of PICRUSt and FUNGuild

Bacterial functional potential was predicted using PICRUSt against the KEGG database, revealing predominant pathways at the primary level, including amino acid metabolism, ABC transporters, and amino sugar and nucleotide sugar metabolism. Among them, African trypanosomiasis is the primary function (6.2% and 5.6%, respectively) in the UHE and UHA samples, while ABC transporters is the primary function (5.1% and 5.2%, respectively) in the HE and HA samples (Fig. 7A).

Figure 7 Function prediction of PICRUSt and FUNGuild.

(A) Relative abundance of predicted KEGG Orthologs functional profiles (KEGG level 1) of bacteria, (B) relative abundance of predicted trophic mode of fungi. Note: HE, UHE represent endophytes in the uninfected and infected leaves of V. unguiculata, respectively; HA, UHA represent the epiphyte in the uninfected and infected leaves of V. unguiculata, respectively.

Functional analysis of the fungal communities was conducted using FUNGuild to determine their trophic modes and functional guilds across samples. The results show that eight trophic mode groups could be classified, including Symbiotroph, Saprotroph-Symbiotroph, Saprotroph-Pathotroph-Symbiotroph, Saprotroph, Pathotroph-Symbiotroph, Pathotroph-Saprotroph-Symbiotroph, Pathotroph-Saprotroph, and Pathotroph. While all of the OTUs that were not matching with any taxa in the database were categorized as “unassigned”. Pathotroph-Saprotroph-Symbiotroph was the dominant trophic mode in HE and HA samples (62.2% and 45.6%, respectively). While the saprotrophic mode was the dominant trophic mode in two samples (UHE, UHA), with relative abundances of 99.5% and 49.1%, respectively after infection of T. urticae. (Fig. 7B).

Assembly processes of phyllosphere-associated microbial communities

Null model analyses revealed that T. urticae infection significantly altered the assembly patterns of phyllosphere bacteria and fungal communities (Fig. 8). Among them, the dominant assembly processes of both endophytic and epiphytic bacterial communities were stochastic processes (HE 91.3% and HA 61.3%, respectively) before T. urticae infection (Figs. 8A, 8C, Table S2), while deterministic processes were dominant assembly processes for endophytic and epiphytic bacterial communities (UHE 51.5% and UHA 76.4%) after T. urticae infection (Figs. 8B, 8D, Table S2). For endophytic and epiphytic fungal communities, deterministic processes were dominant assembly processes (HE 78.2% and HA 77.9%) before T. urticae infection (Fig. 8E, 8G, Table S2), stochastic and deterministic processes jointly governed community assembly, with stochasticity playing a highly significant role (UHE 50.3% and UHA 74.7%) after T. urticae infection (Figs. 8F, 8H, Table S2).

Figure 8 The phyllosphere microbial community assembly based on null model.

(A) Represent the assembly process of endophytic bacteria in the uninfected leaves of V. unguiculata; (B) represent the assembly process of endophytic bacteria in the infected leaves of V. unguiculata; (C) represent the assembly process of epiphytic bacteria in the uninfected leaves of V. unguiculata; (D) represent the assembly process of epiphytic bacteria in the infected leaves of V. unguiculata; (E) represent the assembly process of endophytic fungi in the uninfected leaves of V. unguiculata; (F) represent the assembly process of endophytic fungi in the infected leaves of V. unguiculata; (G) represent the assembly process of epiphytic fungi in the uninfected leaves of V. unguiculata; (H) represent the assembly process of epiphytic fungi in the infected leaves of V. unguiculata. HE, UHE represent endophytes in the uninfected and infected leaves of V. unguiculata, respectively; HA, UHA represent the epiphyte in the uninfected and infected leaves of V. unguiculata, respectively.

Discussion

Plant phyllosphere, as the primary site for photosynthesis, provide organic nutrients that sustain diverse phyllosphere microbial communities. These microbes colonize leaf surfaces by utilizing carbohydrates and organic acids secreted by plants, while dynamically interacting through nutrient competition and mutualistic/inhibitory relationships (Ehau-Taumaunu & Hockett, 2023). Phyllosphere microbes are indispensable in maintaining plant health, enhancing environmental adaptation, and safeguarding ecosystem stability. They inhibit the invasion of pathogenic bacteria, promote plant growth and development, and help plants better adapt to environmental changes (Xu et al., 2022). Recent studies have highlighted how pathogen attacks reshape phyllosphere microbiota, offering new insights into plant-microbe-pathogen tripartite interactions (Díaz-Cruz & Cassone, 2022; Zhang et al., 2018), while pest influences have been less studied. Studies showed that biotic stress can significantly alter plant-associated microbial community composition and diversity (Liu et al., 2025). Cong et al. (2024) reported that alpha diversity and community structure of phyllosphere communities were changed after powdery mildew-induced of cucumber. Rhizosphere microbial communities of verticillium wilt diseased infection of cotton showed lower alpha diversity than healthy plants (Wei et al., 2021). In this study, the microbial alpha diversity of T. urticae-infected plants was significantly lower than that of healthy plants. The results suggest that the infection of pests would lead to lower bacterial and fungal diversity in the phyllosphere. Kong et al. (2016) mentioned that whitefly infection resulted in significant changes in the composition of rhizosphere microbial communities, specific microbial taxa changed in abundance, such as the relative abundance of Pseudomonas increased rapidly after whitefly infection (Kong et al., 2016). In our study, the infection of T. urticae significantly increased the dominant abundance of Proteobacteria (epiphytic) among bacteria (from 84.27% to 93.36%), and decreased the abundance of Firmicutes (epiphytic) in the phyllosphere, decreased from 9.48% to 5.16%. Meanwhile, infection of T. urticae significantly enhanced the abundance of Basidiomycota (endophytic) among fungi, with an increase in abundance from 30.94% to 92.30%, and decreased the abundance of Ascomycota (endophytic) in the phyllosphere, representing a loss of 36.24 percentage points. It is important to note that the 16S/ITS amplicon sequencing data are compositional, meaning that the abundances of all taxa within a sample are expressed as relative proportions summing to 100%. These results were different from the reports of Chen et al. (2020), who found the abundance of Firmicutes was significantly reduced, while Proteobacteria were enriched. This contributed to shaping the bacterial community in leaves and subsequently causing leaf disease. These results indicated that different biotic stress can have different influences on plant-related microbial communities. We speculate that the reason why different biotic stresses have different effects on plant-associated microbial communities is that hosts respond differently to different biotic stresses. Existing studies have demonstrated that insect pest infestation affects the phyllosphere microbiota of plant hosts (Messal et al., 2022; Humphrey & Whiteman, 2020). For example, when plants are infested by pests, they recruit beneficial bacteria, induce the plant host to release certain metabolites, and activate the host’s defense system (Li et al., 2024b). Gao et al. found that when plants are infected by pathogens, they actively recruit beneficial microbes and enhance their own defense ability by promoting the formation of microbial biofilms (Gao et al., 2021).

Previous studies reported that plants can recruit beneficial microorganisms by releasing chemical signals that improve stress resistance (Liu et al., 2020a). In our work, the circos plots showed that the relative abundance changes of some microbial genus after infection, they are defined as newly recruited microbes that contribute to plant defense, such as Rickettsia was significantly increased after infection. This result suggests that Rickettsia has stronger adaptability to the plant physiological changes induced by pest infestation, or has formed a specific interaction with the host plant, enabling it to more easily gain an advantage in the phyllosphere microenvironment after infestation. Existing studies have demonstrated that Rickettsia can promote the synthesis of salicylic acid (SA) by activating the activity of salicylic acid synthase in plants (Shi et al., 2021). As a key signaling molecule in plant defense responses, salicylic acid can further trigger the systemic acquired resistance (SAR) of plants, thereby enhancing the plant’s resistance to pests or pathogens (Zhang, Bouwmeester & Kappers, 2020). Based on this, this study hypothesizes that the significant enrichment of Rickettsia after pest infestation may be the result of “active regulation” by the plant: The plant promotes the proliferation of Rickettsia, and by virtue of Rickettsia’s role in activating salicylic acid synthesis, accelerates the initiation of its own defense mechanisms. Conversely, Rickettsia can also achieve increased abundance by relying on nutrients provided by the plant, forming a mutually beneficial “plant-microbial” interaction.

Studying phyllosphere microbial interactions through co-occurrence network of phyllosphere microbes can offer more insights into how microbiome variations affect plant health. Our study found that connectivity and complexity of co-occurrence network were reduced after infection of T. urticae (Fig. 6). This may represent a “simplification strategy” adopted by the microbial community to maintain basic functions under persistent biotic stress. A network with fewer connections and higher modularity is likely to restrict the spread of disturbances within the community, thereby enhancing its robustness in stressful environments. Coolen et al. demonstrated that Nezara viridula directly inoculates a small number of dominant bacterial species onto plants via its saliva and feces. The introduction of these highly competitive “keystone taxa” is likely to reshape the microbial network. They may outcompete indigenous microorganisms for resources and ecological niches, thereby disrupting the original complex interaction network among indigenous microorganisms and ultimately resulting in a network with fewer connections and a simpler structure. This result is different from the reports of Li et al. (2022), who reported that phyllosphere microbial networks become more complex after Diaporthe citri infection. Thus, we hypothesize that different biotic stresses had different influences on connectivity and complexity of co-occurrence networks of phyllosphere microbes. The core of this difference lies in the differential regulatory effects exerted by the two on the interference patterns of the host microecosystem and the composition of the microbial community structure.

PICRUSt analysis can predict the function of bacterial communities with high dependability (Langille et al., 2013), and it has been used to study many plants-related bacterial functions (Luo et al., 2017). In our study, the results of PICRUSt analysis showed that the “detoxification” ability of ABC transporters in the leaves after infection was reduced, and the substances that were adverse to cell growth could not be discharged from the plant, resulting in leaf wilting. The dominant ABC transporters can “detoxicate” by expelling substances like antibiotics and fatty acids from cells, keeping non-essential foreign substances or secondary metabolites at low levels. This significantly improves the cell survival rate (Qu et al., 2020). Given the limitations of PICRUSt functional prediction analysis, future studies should employ metagenomics to validate and better understand how endophytes and epiphytes influence plant metabolism. FUNGuild is commonly used to compare fungal functions and identify specific functional classifications. The results show that saprotroph were the main functions in infected plants phyllosphere. Saprotrophic fungi obtain C by the decomposition of complex organic matter, which have the capacity to translocate nutrients and to produce multiple enzymes involved in the decomposition of complex organic matter (e.g., cellulose, hemicelluloses, chitin, pectin, and lignin), renders them highly efficient decomposers of bulky substrates, such as litter or dead wood (Crowther, Boddy & Hefin Jones, 2012). Although FUNGuild provides insights into fungal functions, it has limitations due to its reliance on existing literature and data. Therefore, to comprehensively understand the functions of endophytic and epiphytic fungi, we need to further investigate the classification and functional groups of soil fungi. Notwithstanding their constraints, the use of these predictive tools represents a pragmatic choice for the initial assessment of functional potential across a large number of samples, offering insights that guide targeted meta-omics or experimental validation.

In our work, the null model analysis indicates that assembly of epiphytic and endophytic fungal community changed from deterministic processes to stochastic processes after infection of T. urticae, while assembly of epiphytic and endophytic bacterial community changed from stochastic processes to deterministic processes. It is worth noting that the response of community assembly mechanisms to stressors is not an isolated phenomenon. Liu et al. (2025) mentioned that different stresses (e.g., drought, salinity, and disease) affect the structure and function of microbial communities by altering their assembly mechanisms. This shift of assembly mechanism implied that pest infection, as a typical biotic stressor, may have reshaped the rules of microbial community assembly.

Conclusions

Overall, we provide novel evidence that T. urticae infection significantly changed the diversity, community composition, co-occurrence network and function of the phyllosphere microbes. After infection of T. urticae, V. unguiculata recruited some beneficial microbes to the phyllosphere, and changed the assembly of phyllosphere microbial community. Our findings provided a new insight into interactions among phyllosphere microbes-pest-plant.

Supplemental Information

Supplemental Information 1 Topological properties of microbial networks

a: The number of connections obtained by SparCC correlations. b: Microbial taxon (at ASV level) with at least one significant (P < 0.05) and strong (SparCC > 0.7 or < −0.7) correlation c: SparCC positive correlation (> 0.7 with P < 0.05) d: SparCC negative correlation (< −0.7 with P < 0.05) e: The node connectivity, that is, the average number of connections per node in the network. HE, UHE represent e ndophytes in the uninfected and infected leaves of V. unguiculata, respectively; HA, UHA represent the epiphyte in the uninfected and infected leaves of V. unguiculata, respectively.

Supplemental Information 2 Relative contribution of components in the assembly process of bacterial and fungal communities

The sum of the proportions of Deterministic (%) and Stochastic (%) is 100%. Whichever process exceeds 50% for a sample is the dominant process for that sample. HE, UHE represent endophytes in the uninfected and infected leaves of V. unguiculata, respectively; HA, UHA represent the epiphyte in the uninfected and infected leaves of V. unguiculata, respectively.

Supplemental Information 3 Morphological Characteristics of Vigna unguiculata Infected by Tetranychus urticae

Note: (a) and (b) uninfected phyllosphere of Vigna unguiculata, (c) phyllosphere of Vigna unguiculata infected by Tetranychus urticae, (d) leaf morphology of Vigna unguiculata after infection.

Supplemental Information 4 The influence of T. urticae on the composition of bacteria and fungi in both infected and uninfected leaves

(a) the dominant phyllosphere bacterial genera, (b) the dominant phyllosphere fungal genera. The left half circle represents the dominant genera, and proportions of each genus in different samples. The right half circle represents the different samples. HE, UHE represent endophytes in the uninfected and infected leaves of V. unguiculata, respectively; HA, UHA represent the epiphyte in the uninfected and infected leaves of V. unguiculata, respectively.

Additional Information and Declarations

Competing Interests

Author Contributions

Data Availability

The authors declare there are no competing interests.

DaWei Chen conceived and designed the experiments, performed the experiments, analyzed the data, prepared figures and/or tables, and approved the final draft.

GaoQin Xia conceived and designed the experiments, performed the experiments, analyzed the data, prepared figures and/or tables, and approved the final draft.

JiaoJiao Wang conceived and designed the experiments, authored or reviewed drafts of the article, and approved the final draft.

YanYan Luo conceived and designed the experiments, authored or reviewed drafts of the article, and approved the final draft.

HongLi Wang conceived and designed the experiments, authored or reviewed drafts of the article, and approved the final draft.

Jing Zhao conceived and designed the experiments, authored or reviewed drafts of the article, and approved the final draft.

Kun Sun conceived and designed the experiments, authored or reviewed drafts of the article, and approved the final draft.

The following information was supplied regarding data availability:

The raw sequence data are available at NCBI BioProject: PRJNA1026786.

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
