# Peer review of "Effects of Tetranychus urticae infection on phyllosphere microbial community assembly of Vigna unguiculata"

_PeerJ, doi:10.7717/peerj.20389_

## Round 0.1 · original submission · Major Revisions

· Academic Editor

Major Revisions

Three experts assessed your manuscript and found merit in the content. There are some issues with the form and questions about the experimental design that need to be addressed before moving to the next stage of the editorial process.

**Language Note:** The review process has identified that the English language must be improved. PeerJ can provide language editing services - please contact us at [email protected] for pricing (be sure to provide your manuscript number and title). Alternatively, you should make your own arrangements to improve the language quality and provide details in your response letter. – PeerJ Staff

·

Basic reporting

Dear author/authors;
Your study examining the effect of the two-spotted spider mite infection on cowpea on phyllosphere bacterial load is likely to make significant contributions to the literature. Your work is suitable for publication after the deficiencies/errors noted in the attached file are corrected. I hope that in your future work, you will investigate the infection of species that can alter the density and diversity of phyllosphere bacteria in a way that benefits plants.
Best regards.

Abstract
Line 22: The Oxford comma is used throughout the article. A similar method should be used throughout the manuscript.
Keywords
Lines 29-30: The words used in the title should not be used in the keywords. Additionally, the keywords should be listed in alphabetical order.
Introduction
The introduction is well written and summarizes the background of the topic.
Lines 61: The study cited as Mendes et al., 2018 is not a study on Arabidopsis thaliana, contrary to what is mentioned in the sentence.

Experimental design

Materials & Methods
The Materials and Methods section is generally well written, with a few technical errors, and is detailed enough to allow for reproducibility of the study. Additionally, the meanings of HE, HA, UHE, and UHA must be specified in the material methods section. Additionally, this information should be included at the bottom of the relevant figures and tables.
Line 79: The first letters of the words must be capitalized.
Line 85: Need a gap between “0.5” and “g”.
Lines 93 and 97: Is the method belong to them? If not, please refer the original paper.
Lines 105 and 106: Is the method belong to them? If not, please refer the original paper.
Lines 109 and 118: In my opinion, the method and the reference Penton et al., 2016 seem to have no relevance. Please check.
Line 125: In my opinion, the method and the reference Grady et al., 2019 seem to have no relevance. Please check.
Line 133: In my opinion, the method and the reference Li et al., 2022 seem to have no relevance. Please check.
Line 135: Is the method belong to them? If not, please refer the original paper.

Validity of the findings

Results
The results section is generally well-written and summarizes the study's findings. Detected errors are listed below.
Line 161: The sentence must start with a capital letter
Lines 161-163: The sentence is not understandable. It should be rewritten to convey the intended meaning.
Line 169: The same word was used twice. It could be written as (...increased (92.21%).).
Lines 201 and 205: I think the word “infestion” should be “infection”.

Discussions
The discussion section of the study is generally well-written and effectively explores the topic within the literature. Detected shortcomings are listed below.
Line 254: The citation (Diaz-Cruz et al., 2022) is not in references section.
Lines 256-257: Although many studies were mentioned in the sentence, only one study was cited.
Lines 266-271: For better readability, the sentence can be changed as follows (...abundance of Proteobacteria (epiphytic), ....abundance of Firmicutes (epiphytic), ... abundance of Bacidiomycota (endophytic)).
Lines 276-277: Although many studies were mentioned in the sentence, only one study was cited.
Line 313: Need a gap after end of sentence.
Line 314: Need a dot after “..et al”.

Additional comments

References
Please use same type of reference writing rule. Among the references, there are examples of journal names written in full, as well as examples of shortened versions.
Line 341: Short journal name.
Lines 348 and 351: The references are written same in manuscript (Chen et al., 2020). You should add “a” and “b” after them. (Chen et al., 2020a and Chen et al., 2020b)
Line 358: Short journal name.
Line 361: Short journal name.
Line 364: Short journal name.
Line 367: Short journal name.
Line 370: Short journal name.
Line 380: Short journal name.
Line 383: Short journal name.
Line 386: Short journal name.
Line 407: First author surname not same in manuscript.

Reviewer 2 ·

Basic reporting

The manuscript is generally well-written in professional language, and addressing a knowledge gap in pest-microbe-plant interactions,. Figures/tables are well-prepared with suitable label. Raw data is also provided.

Experimental design

The study addresses a clear and relevant research question on pest–microbe–plant interactions, fitting well within the journal’s scope. The methods—covering sampling, DNA extraction, sequencing, and bioinformatics—are described in adequate detail for replication, and the combination of diversity, network, functional, and null model analyses provides a robust framework.

Minor clarifications are recommended: (1) specify the number of plants per replicate and whether replicates were pooled; (2) confirm that infected and uninfected samples were collected under comparable environmental conditions; (3) provide thresholds used for network construction; and (4) briefly justify the use of PICRUSt and FUNGuild despite their known limitations.

Validity of the findings

NA

Additional comments

Before acceptance, I recommend the authors:

Conduct a careful language and formatting check to correct minor errors and inconsistencies.

Clarify sampling details to ensure reproducibility.

Slightly expand figure legends to better guide interpretation.

Reviewer 3 ·

Basic reporting

a) on format

The English language should be improved to ensure that the text is understandable. Some examples where the sentences should be rewritten include lines 35, 39-41, 133-135 and through the whole manuscript.

The structure of some paragraphs should be reviewed, for example in the introduction. It is difficult to understand the link between the references from studies conducted on the phyllosphere and on the rhizosphere etc.

Some references in the Literature references should be checked. As examples, Chen et al. (2020) with two references, Diaz-Cruz and Cassone and not Gustavo A Diaz-Cruz…, Mago is incomplete, etc.

Typing should be reviewed and spaces corrected (epiphyt ic, epiphyt e etc.).

Some figures could be summarized in the text as the results are presented without further discussion (for example, Figure 4).

b) on content

The article should include sufficient background information on how biotic stress generated by insect pests may have an impact on the diversity and structure of microbial communities associated with host plants. Almost all references in the article relate to pathogen attacks: there is only two references on Tetranychus and one on whiteflies.

As pathogens and pests (insects, mites etc.) interact with their hosts very differently (and plants may respond differently), it is expected that the impact on the microbial communities will be quite different. As a consequence, what are the questions that are addressed and the hypotheses that justify the analyses?

The manuscript takes into account the endophytic and epiphytic bacteria and fungi encountered in and on the leaves of Vigna unguiculata. What are the underlying hypotheses that justify the choice of V. unguiculata-Tetranychus urticae as study system?

Experimental design

The main objective of the study is to compare the microbial communities encountered in Tetranychus urticae-infected or uninfected leaves of V. unguiculata. To be able to understand the potential mechanisms that explain the differences between “infected” and “uninfected” leaves, important information is missing on the origin of the leaf samples that were used for the characterization of the microbial communities:

- were the leaves with or without Tetranychus urticae, or with or without symptoms due to past attack by the spider mite? How long were the spider mites present on the host plant?

- were the damaged and undamaged leaves collected on the same plant or on different plants that were grown under the same abiotic conditions?

- were the three biological replicates collected on three different plants?

Statistical analyses should be justified.

After an attack by T. urticae, induction of direct and/or indirect plant defenses is very often time-dependent and could play a role in the microbial community assembly.

Validity of the findings

Presentation of the results should be reviewed :

- Some statements are confusing. When differences in diversity indices are not statistically significant, this should be mentioned (lines 138-144), 50.3% does not mean that stochastic processes were dominant (line 242).

- Are the different results consistent? In the co-occurrence network analysis, does the diversity of the different communities have an impact on the conclusions on connectivity and complexity?

- Are the microbial functional groups reliable? The functional classification of the bacteria and fungi encountered on the leaves of V. unguiculata seems very questionable. For example, it is known that for some fungal genera, one genus can be a saprophyte or a pathogen depending on the context…

- What is the link with soil fungi (line 309)?

- In the discussion (lines 267-270), it should be noted that relative abundances are given and a decrease in one group translates into an increase in another group.

The main conclusion given by the authors is that “Results indicated that different biotic stresses can have different influences on plant-related microbial communities”. This is expected. The question is why and which bacteria and fungi that are the most impacted. Would it be possible to get more insights into the underlying mechanisms (or at least hypotheses)? Previous studies have compared microbial communities between plants that were attacked by pests (whitefly etc.). The discussion (and introduction) should be rewritten to better discuss the consequences that pests may have on microbial communities.

Additional comments

As mentioned in the manuscript, studies of the impact of biotic stress on microbial communities have mostly considered pathogens. This study clearly fills a gap. However, statements as to how the study contributes to filling it are needed, in particular, the relevance and main contribution of the results to our understanding of plant-pest-microbiota interactions.

---

## Round 0.2 · Minor Revisions

· Academic Editor

Minor Revisions

The Reviewers appreciated the changes in the revised version of the manuscript, but there are still some minor issues to attend. Please address them in a third version of the manuscript.

·

Basic reporting

Dear author(s),

your work contributes to the literature. It appears that you have revised the sections mentioned in the previous review. Thank you for your efforts. However, I believe you have overlooked a point mentioned below in the method section. I kindly ask you to consider this if a similar situation exists throughout the article.

Best regards.

Experimental design

Lines 106-108: I don't know which method you used. However, this article is a method publication, about the vacuum filtration method you used (Isolation of bacteriophages from air using vacuum filtration technique: an improved and novel method | Journal of Applied Microbiology | Oxford Academic). You should cite the relevant publication of the researchers who developed your method, not other studies that used it. This way, anyone reading your article who wants to use the same method can easily find the relevant publication and apply it as you did. Furthermore, imagine you developed that method yourself. And people aren't citing your work, but rather citing someone who has done research using the method you developed.

Validity of the findings

There is no problem.

Reviewer 2 ·

Basic reporting

The revision version already addressed all comments from reviewers.

Experimental design

The experimental design is clear and supported by the results.

Validity of the findings

The figures/tables in this revision manuscript can support the findings authors draw well.

Additional comments

NA

Reviewer 3 ·

Basic reporting

In the revised version, the authors have addressed most questions on format (English -still some small mistakes remain in the document-, references – there are two references Liu et al. 2020 etc.-).

In the introduction, no additional reference has been added and all references are related to pathogen attacks either in the rhizosphere or the phyllosphere? As pathogens and pests (insects, mites etc.) interact with their hosts very differently (and plants may respond differently), it is expected that the impact on the microbial communities will be quite different. As a consequence, what are the hypotheses that justify the analyses and could explain the expected impacts?

No additional reference has been included to justify the choice of V. unguiculata-Tetranychus urticae as study system besides the fact that there are both very common. Is there any study on V. unguiculata defense system? Or induction by Tetranychus ?

Experimental design

Presentation of the experimental design has been greatly improved. However, is there any data to justify the fact that infestation with spider mites lasted 15 days (see above)? After an attack by T. urticae, induction of plant defenses is very often time-dependent and could play a role in the dynamics of microbial community assembly.

Validity of the findings

Presentation of the results has been revised and this new version addresses most issues.

Li et al. (2024) has been added as a reference on the consequences of pest attacks on plant microbial community. If possible, more references would have provided new elements for comparison with this study on the impacts of pests on plant microbiota.

Additional comments

As mentioned in the manuscript, studies of the impact of biotic stress on microbial communities have mostly considered pathogens. This study clearly fills a gap and should emphasize how different biotic stresses may have different impacts on connectivity and complexity of co-occurrence networks with the phyllosphere microbial community.

---

## Round 0.3 · accepted · Accept

· Academic Editor

Accept

The authors have addressed all the comments raised by the Reviewers. It is now suitable for moving on to the next editorial stage.